# Morphological and Physiological Response of Different Lettuce Genotypes to Salt Stress

Bikash Adhikari, Omolayo J. Olorunwa, Jeff C. Wilson and T. Casey Barickman *

Department of Plant and Soil Sciences, North Mississippi Research and Extension Center, Mississippi State University, Verona, MS 38879, USA; ba917@msstate.edu (B.A.); omo26@msstate.edu (O.J.O.); jeff.wilson@msstate.edu (J.C.W.)
* Correspondence: t.c.barickman@msstate.edu; Tel.: +1-(662)-566-2201

**Abstract:** Salt stress (SS) refers to excessive soluble salt concentrations in the plant root zone. SS also causes cellular water deficits, ion toxicity, and oxidative stress in plants, all of which can cause growth inhibition, molecular damage, and even plant mortality. Lettuce (*Lactuca sativa* L.) has a threshold electrical conductivity of 1.3–2.0 dS/m. Thus, this research focused on physiological, morphological, and biochemical attributes in multiple lettuce genotypes under SS compared to plants grown under control conditions. The experiment was arranged in a randomized complete block design with four replications. One month after planting, the salt treatment was applied at the rate of 100 millimoles (mM). The 0 mM salt in water treatment was considered the control. A significant effect of SS on different morphological and physiological traits was observed in one-month-old lettuce plants. PI 212099, Buttercrunch-1, and PI 171676 were highly salt-tolerant. Genotypes with high salt tolerance usually had poor growth potential under control conditions. This suggests that the morphological and physiological response of 38 lettuce cultivars towards SS is genotype dependent. Identifying SS's physiological, morphological, and biochemical attributes in lettuce may help plant-breeders develop salt-tolerant lettuce genotypes.

**Keywords:** salt tolerance; leaf number; salt sensitivity; butterhead; leafy; crisphead; romaine

## 1. Introduction

The changing progression of climate is a significant factor in the global agricultural system [1]. Salt stress (SS) is abiotic stress that significantly limits agricultural production worldwide [2]. Plants naturally accumulate salts at particular levels that are not hazardous in general, with sodium chloride (NaCl) concentration at specific levels working as a eustressor, helping improve crop quality [3]. However, no toxic substance is more pervasive in limiting plant growth than a higher concentration of salts [4]. Zu [4] also reported that the stress created by higher salt concentrations is two-fold, especially in the soil solution. High salt concentration, or SS, refers to excessive soluble salts in the plant root zone, making it difficult to extract the soil's water and essential mineral nutrients [5]. It affects the plant in two different ways: (1) it causes osmotic stress by lowering soil water potential, restricting water intake, and (2) it causes excessive ion absorption, notably, $Na^+$ and Cl, which interferes with numerous metabolic processes [6]. High salt concentrations affect 100 million hectares (5% of arable land) and 45 million hectares (20% of irrigated land) around the world, limiting crop growth and annual crop output [7,8]. Each year, salt accumulation is predicted to damage 10 million hectares of agricultural land worldwide [9]. Since most crops are glycophytic, there is an evident negative correlation between agricultural production and SS [10]. SS also causes cellular water deficits, ion toxicity, and oxidative stress in plants, all of which can cause growth inhibition, molecular damage, and even plant mortality [11]. Therefore, understanding the absorption processes, movement within plants, and translocating major toxic ions (especially $Na^+$ and $Cl^-$) is critical for improving salt tolerance in crops.

The sensitivity of the crop to SS varies from one crop to another [12]. Being a moderately salt-sensitive crop, lettuce (*Lactuca sativa* L.) has a threshold electrical conductivity of 1.3–2.0 dS/m; this salt level is considered beneficial for most lettuce varieties and above that deemed detrimental [13]. Lettuce is a widely consumed salad vegetable in the United States [14], and it accounts for around 22% of total lettuce production worldwide [15]. Lettuce contains various polyphenols, such as flavonoids and anthocyanins, responsible for antioxidant activity [16,17]. However, several studies have reported that higher salt concentration inhibits lettuce germination, leaf water content, photosynthetic rate, chlorophyll content, root and shoot growth, and polyphenolic compounds [18–20].

In general, most of the crops, except for halophytes will not thrive well under the high concentrations of NaCl salt of about 400 mM [21]. Besides, most of the lettuce cultivars are glycophyte that are severely inhibited under 100–200 mM NaCl [4]. C3 plants such as lettuce are usually cultivated under a wide range of well-drained soil conditions. However, the SS condition is unsuitable for most of the lettuce cultivars as it induces oxidative stress and can damage the plant's physiological attributes [6,22–24]. Similarly, there are very few studies on screening lettuce cultivars under SS, and most PIs have not been screened for salt tolerance yet. Therefore, there is a need to identify the unidentified salt-stress lettuce genotypes which can grow under SS in order to study their physiological and morphological attributes under SS. Another study approach is to help breeders develop salt-tolerant lettuce cultivars through trials based on intraspecific variation. The potential of SS tolerance and the economic value of lettuce in the US market makes it an excellent commercial crop for growers, even in arid and semi-arid regions. Thus, this research focused on identifying physiological, morphological, and biochemical attributes in multiple lettuce genotypes under SS compared to plants grown under control conditions.

## 2. Results

### 2.1. Morphological Traits

2.1.1. Leaf Number

Butterhead-type (BH) lettuce genotypes' leaf number (LN) was significantly affected by salt treatment, except for Buttercrunch-2 and Plant Introduction 342448 (PI 342448) (Table 1) compared to control. There was a significant reduction in LN from 2.5% (PI 615052) up to 14% (PI 342515) under SS compared to control. In crisphead-type lettuce (CH), LN of PI 593426 was significantly reduced by 15.4%, while LN of PI 635077 increased significantly by 9.3% compared to control. Almost all the leaf-type lettuces (LT) were affected by SS as there was a significant reduction in LN from 3% (PI 171676) to 30% (PI 171675) compared to control (Table 2). LN of PI 212099 (LT) remained unaffected (1% increase) under SS compared to control. LN of Romaine-type (RT) lettuce (PI 536708 and PI 612664) was increased significantly by 10.9% and 15.3%, respectively, under SS compared to control. On the other hand, LN of Green Forest and PI 278066 (RT) were significantly reduced by 12.8% and 4.9% under SS compared to control. However, PI 274366 (RT) remained unaffected as there was no significant change in the LN under SS compared to control.

**Table 1.** Leaf number (LN) (mean ± standard deviation, $n = 6$) of butterhead- and crisphead-type lettuce cultivars grown under control and salt-stress conditions (100 mM) after 42 days of sowing (10 days after salt treatment).

| Type | Variety | LN [1] | | | | | | | % [3] |
|---|---|---|---|---|---|---|---|---|---|
| | | Control | | | | Salt | | | |
| Butterhead | Burpee Bibb | 12.2 | ± | 1.4 | a [2] | 10.9 | ± | 1.6 | a | −10.7 |
| | Buttercrunch-1 | 12.7 | ± | 1.8 | a | 12.4 | ± | 3.4 | a | −2.4 |
| | Buttercrunch-2 | 12.9 | ± | 3.6 | a | 12.9 | ± | 2.1 | a | 0.0 |
| | Hybrid Bibb | 15.5 | ± | 2 | a | 12.3 | ± | 0.3 | a | −20.6 |
| | PI 273606 | 11.9 | ± | 1 | a | 11.3 | ± | 2.3 | a | −5.0 |
| | PI 342448 | 10.8 | ± | 1 | ab | 11.6 | ± | 1.7 | a | 7.4 |
| | PI 342515 | 14.7 | ± | 0.9 | a | 12.6 | ± | 0.8 | a | −14.3 |
| | PI 358020 | 12.7 | ± | 1.5 | a | 12.2 | ± | 1.8 | a | −3.9 |
| | PI 372908 | 12.8 | ± | 2 | a | 11.8 | ± | 2 | a | −7.8 |
| | PI 615052 | 11.8 | ± | 2 | a | 11.5 | ± | 1.6 | a | −2.5 |
| | PI 634671 | 12.1 | ± | 1.3 | a | 10.8 | ± | 2.3 | ab | −10.7 |
| Crisphead | Crispino | 8.4 | ± | 1.4 | hijk | 8.2 | ± | 2.7 | jk | −2.4 |
| | Iceberg | 10.3 | ± | 0.2 | abc | 10 | ± | 1.9 | abcde | −2.9 |
| | Parris Island | 9.9 | ± | 0.9 | abcde | 10.1 | ± | 2.2 | abcd | 2.02 |
| | PI 177423 | 10.2 | ± | 2 | abcd | 9.1 | ± | 1.2 | abcd | −11 |
| | PI 536803 | 8.1 | ± | 1.8 | k | 7.6 | ± | 1.8 | k | −6.2 |
| | PI 536822 | 8.5 | ± | 2 | ghijk | 8.3 | ± | 3.3 | jk | −2.4 |
| | PI 593426 | 9.1 | ± | 1.6 | cdefgh | 7.7 | ± | 0.8 | k | −15 |
| | PI 600773 | 9.4 | ± | 0.5 | bcdefg | 9.2 | ± | 1.2 | abcdef | −2.1 |
| | PI 635075 | 9.1 | ± | 2.2 | cdefgh | 8.9 | ± | 1.2 | defghi | −2.2 |
| | PI 635077 | 8.6 | ± | 2.9 | abcdef | 9.4 | ± | 1.4 | cdefg | 9.3 |
| | Prizehead | 10.7 | ± | 2.2 | ab | 10.0 | ± | 1.2 | abcde | −6.5 |

[1] Leaf number on per plant basis. [2] Means followed by different letters are significantly different from each other ($p < 0.05$) as assessed by Tukey's test. [3] % indicates the percentage change in each variety due to change in treatment levels where negative sign (−) indicates a percentage decrease in LN.

**Table 2.** Leaf number (LN) (mean ± standard deviation, $n = 6$) of leaf- and romaine-type lettuce cultivars grown under control and salt-stress conditions (100 mM) after 42 days of sowing (10 days after salt treatment).

| Type | Variety | LN [1] | | | | | | | % [3] |
|---|---|---|---|---|---|---|---|---|---|
| | | Control | | | | Salt | | | |
| Leaf | PI 171675 | 11.4 | ± | 1.8 | a [2] | 8 | ± | 0.7 | k | −30 |
| | PI 171676 | 8 | ± | 2.1 | k | 7.8 | ± | 1.2 | k | −2.5 |
| | PI 175737 | 9.6 | ± | 0.6 | abcde | 8 | ± | 1 | k | −17 |
| | PI 212099 | 9.5 | ± | 2.4 | abcde | 9.6 | ± | 1.9 | abcde | 1.05 |
| | PI 358018 | 13.3 | ± | 2.6 | a | 10.9 | ± | 1.4 | ab | −18 |
| | PI 601339 | 10.6 | ± | 1.1 | ab | 9.4 | ± | 1.7 | abcde | −11 |
| | PI 601488 | 10.3 | ± | 3 | abcd | 9.3 | ± | 2.3 | bcdef | −9.7 |
| | Ruby | 9.4 | ± | 1.4 | bcdef | 7.6 | ± | 2.1 | k | −19 |
| Romaine | Burgundy Delight | 8.8 | ± | 2.3 | efghij | 9.4 | ± | 0.3 | abcde | 6.82 |
| | Green Forest | 10.9 | ± | 0.8 | a | 9.5 | ± | 1.5 | abcde | −13 |
| | PI 278066 | 10.2 | ± | 2.4 | abcd | 9.7 | ± | 1.8 | abcde | −4.9 |
| | PI 289023 | 11.6 | ± | 0.6 | a | 12.4 | ± | 1.6 | a | 6.9 |
| | PI 536708 | 11.0 | ± | 1.4 | a | 12.2 | ± | 1.2 | a | 10.9 |
| | PI 612664 | 9.8 | ± | 0.8 | abcde | 11.3 | ± | 1.0 | a | 15.3 |
| | PI 613577 | 11.3 | ± | 2 | a | 8.3 | ± | 1.3 | ijk | −27 |
| | PI 274366 | 10.3 | ± | 1.6 | abc | 10.2 | ± | 0.9 | abcd | −1 |

[1] Leaf number on per plant basis. [2] Means followed by different letters are significantly different ($p < 0.05$) as assessed by Tukey's test. [3] % indicates the percentage change in each variety due to change in treatment levels where negative sign (−) indicates a percentage decrease in LN.

### 2.1.2. Fresh Mass (FM)

There was a significant increase in FM in six out of eleven BH, increasing from 12.17% (PI 342515) to up to 59.9% (Buttercrunch-1) compared to control (Table 3). On the other hand, PI 615052, PI 273606, and Hybrid Bibb were significantly reduced by 24.3%, 23.8%, and 25.9%, respectively, under SS compared to control. Parris Island (CH) was found highly salt-sensitive with a significant increase (67%) in FM under SS compared to control. On the contrary, PI 536803, Iceberg, and PI 635075 were moderately affected under SS with a significant decrease from 5% to 12.6% compared to control. PI 536822 was highly affected under SS (29.4% decrease in FM). In two out of eight LT lettuces (PI 171676 and PI 212099), FM significantly increased by 104% and 197%, respectively, under SS compared to control (Table 4). On the contrary, PI 171675, PI 601488, and Ruby were found sensitive to SS with a decrease in FM from 35% to 70% compared to control. Burgundy Delight, PI 612664, and PI 536708 showed a significant increase in FM under SS ranging from 4% to 47.8% compared to control, with Burgundy delight showing the highest percentage increase in FM (48%). FM of PI 536708 and PI 289023 were found unaltered by salt treatment. However, Green Forest and PI 278066 showed a significant reduction in FM by 30% and 33%, respectively, when subjected to SS compared to control.

### 2.1.3. Dry Mass (DM)

In 3 out of 11 BT lettuces (PI 615052, Buttercrunch-1, and PI 615052), DM was significantly greater (52%, 34%, and 33%, respectively) when treated with salt to control, as shown in Table 5. Conversely, PI 634671's DM remained unaffected when treated with salt compared to control. Burpee Bibb, Hybrid Bibb, PI 372908, PI 273606, and PI 342515 were found the most salt-sensitive as there was a significant decrease in FM by 35%, 31%, 28%, 27%, and 19%, respectively.

**Table 3.** Fresh mass (FM) (mean $\pm$ standard deviation, $n$ = 6) of butterhead- and crisphead-type lettuce cultivars grown under control and salt-stress conditions (100 mM) after 42 days of sowing (10 days after salt treatment).

| Type | Variety | FM [1] | | | | | | | |
|---|---|---|---|---|---|---|---|---|---|
| | | Control | | | | Salt | | | % [3] |
| Butterhead | Burpee Bibb | 58.8 | $\pm$ | 15 | abcde [2] | 54.3 | $\pm$ | 20 | abcde | −7.7 |
| | Buttercrunch-1 | 39.9 | $\pm$ | 21 | efg | 63.8 | $\pm$ | 31 | abcd | 59.9 |
| | Buttercrunch-2 | 49.5 | $\pm$ | 34 | abcde | 68 | $\pm$ | 13 | abc | 37.4 |
| | Hybrid Bibb | 61.4 | $\pm$ | 38 | abc | 45.5 | $\pm$ | 12 | cde | −26 |
| | PI 273606 | 43.6 | $\pm$ | 25 | cde | 33.2 | $\pm$ | 22 | def | −24 |
| | PI 342448 | 12.6 | $\pm$ | 19 | fg | 14.3 | $\pm$ | 24 | efd | 13.5 |
| | PI 342515 | 41.9 | $\pm$ | 24 | cde | 47 | $\pm$ | 23 | bcde | 12.2 |
| | PI 358020 | 33.5 | $\pm$ | 30 | abcde | 45.3 | $\pm$ | 32 | abcde | 35.2 |
| | PI 372908 | 44.4 | $\pm$ | 19 | abcde | 37.8 | $\pm$ | 21 | abcde | −15 |
| | PI 615052 | 19.3 | $\pm$ | 21 | cde | 14.6 | $\pm$ | 49 | cdef | −24 |
| | PI 634671 | 25.1 | $\pm$ | 10 | bcde | 30.2 | $\pm$ | 25 | bcde | 20.3 |
| Crisphead | Crispino | 53.4 | $\pm$ | 5.5 | ab | 56.7 | $\pm$ | 9.3 | abc | 6.18 |
| | Iceberg | 45.5 | $\pm$ | 15 | abcd | 43.2 | $\pm$ | 20 | abcde | −5.1 |
| | Parris Island | 33.2 | $\pm$ | 26 | bcd | 55.4 | $\pm$ | 56 | abcde | 66.9 |
| | PI 177423 | 33.8 | $\pm$ | 52 | abcde | 40.3 | $\pm$ | 13 | abcd | 19.2 |
| | PI 536803 | 47.1 | $\pm$ | 22 | abcd | 42.5 | $\pm$ | 23 | abcde | −9.8 |
| | PI 536822 | 47.2 | $\pm$ | 16 | abcd | 33.3 | $\pm$ | 26 | abcde | −29 |
| | PI 593426 | 65.3 | $\pm$ | 19 | ab | 60.1 | $\pm$ | 14 | ab | −8 |
| | PI 600773 | 52.9 | $\pm$ | 18 | abc | 63.1 | $\pm$ | 40 | abcd | 19.3 |
| | PI 635075 | 54.1 | $\pm$ | 20 | abcd | 47.3 | $\pm$ | 9.9 | bcde | −13 |
| | PI 635077 | 54.6 | $\pm$ | 9.6 | abcd | 74.9 | $\pm$ | 11 | a | 37.2 |
| | Prizehead | 44.1 | $\pm$ | 5.3 | bcd | 47.0 | $\pm$ | 27 | abcde | 6.58 |

[1] Fresh mass (in gram) on per plant basis. [2] Means followed by different letters are significantly different from each other ($p < 0.05$) as assessed by Tukey's test. [3] % indicates the percentage change in each variety due to change in treatment levels (Control vs. Salt), where negative sign (−) indicates a percentage decrease in FM.

**Table 4.** Fresh mass (FM) (mean ± standard deviation, *n* = 6) of leaf- and romaine-type lettuce cultivars grown under control and salt-stress conditions (100 mM) after 42 days of sowing (10 days after salt treatment).

| Type | Variety | FM [1] | | | | | | | |
|------|---------|--------|---|---|---|---|---|---|---|
| | | Control | | | | Salt | | | % [3] |
| Leaf | PI 171675 | 71.6 | ± | 28 | ab [2] | 20.2 | ± | 5.5 | f | −72 |
| | PI 171676 | 15 | ± | 23 | def | 30.6 | ± | 21 | cdef | 104 |
| | PI 175737 | 28.9 | ± | 13 | cdef | 30 | ± | 2.3 | cde | 3.81 |
| | PI 212099 | 26 | ± | 29 | cdef | 77.1 | ± | 23 | a | 197 |
| | PI 358018 | 39.6 | ± | 33 | abcd | 33.9 | ± | 11 | cde | −14 |
| | PI 601339 | 37.3 | ± | 15 | de | 34.7 | ± | 6.5 | cde | −7 |
| | PI 601488 | 51.9 | ± | 25 | abcde | 31.7 | ± | 14 | cde | −39 |
| | Ruby | 67.4 | ± | 6.9 | a | 43.7 | ± | 17 | bcde | −35 |
| Romaine | Burgundy Delight | 29.9 | ± | 31 | bcdef | 44.2 | ± | 26 | abcde | 47.8 |
| | Green Forest | 62.6 | ± | 33 | abcde | 44 | ± | 22 | abcde | −30 |
| | PI 278066 | 54.5 | ± | 24 | abcde | 36.5 | ± | 16 | cdef | −33 |
| | PI 289023 | 40.6 | ± | 5.5 | bcde | 39.8 | ± | 9.1 | cdef | −2 |
| | PI 536708 | 62.8 | ± | 11 | abc | 65.7 | ± | 3.6 | a | 4.62 |
| | PI 612664 | 30.1 | ± | 9.5 | de | 34 | ± | 11 | cdef | 13 |
| | PI 613577 | 45.5 | ± | 18 | cde | 38.6 | ± | 13 | cdef | −15 |
| | PI 274366 | 32.6 | ± | 18 | cdef | 25.3 | ± | 6.4 | ef | −22 |

[1] Fresh mass (in gram) on per plant basis. [2] Means followed by different letters are significantly different from each other (*p* < 0.05) as assessed by Tukey's test. [3] % indicates the percentage change in each variety due to change in treatment levels (Control vs. Salt), where negative sign (−) indicates a percentage decrease in FM.

**Table 5.** Dry mass (DM) (mean ± standard deviation, *n* = 6) of butterhead- and crisphead-type lettuce cultivars grown under control and salt-stress conditions (100 mM) after 42 days of sowing (10 days after salt treatment).

| Type | Variety | DM [1] | | | | | | | |
|------|---------|--------|---|---|---|---|---|---|---|
| | | Control | | | | Salt | | | % [3] |
| Butterhead | Burpee Bibb | 7.5 | ± | 2.8 | abc [2] | 4.9 | ± | 1.4 | cdef | −35 |
| | Buttercrunch-1 | 6.4 | ± | 1.8 | bcd | 8.6 | ± | 2.9 | abc | 34.4 |
| | Buttercrunch-2 | 7.2 | ± | 4.5 | abcde | 7.8 | ± | 1 | bcde | 8.33 |
| | Hybrid Bibb | 9.5 | ± | 2.8 | A | 6.6 | ± | 1.1 | cd | −31 |
| | PI 273606 | 5.2 | ± | 3.1 | de | 3.8 | ± | 2.8 | ef | −27 |
| | PI 342448 | 1.6 | ± | 1.8 | F | 1.5 | ± | 2.5 | f | −6.3 |
| | PI 342515 | 5.2 | ± | 2.9 | def | 4.2 | ± | 3.4 | def | −19 |
| | PI 358020 | 4.5 | ± | 3.4 | def | 6 | ± | 4.5 | abcde | 33.3 |
| | PI 372908 | 5.4 | ± | 2.2 | def | 3.9 | ± | 2.6 | def | −28 |
| | PI 615052 | 2.3 | ± | 2.1 | ef | 3.5 | ± | 3.7 | def | 52.2 |
| | PI 634671 | 2.8 | ± | 1.4 | ef | 2.9 | ± | 4.3 | def | 3.57 |
| Crisphead | Crispino | 7.5 | ± | 0.8 | ab | 7.7 | ± | 1.1 | ab | 2.67 |
| | Iceberg | 6.6 | ± | 1.7 | abc | 6.1 | ± | 1.3 | bc | −7.6 |
| | Parris Island | 6.1 | ± | 2.6 | abc | 8 | ± | 3.7 | abcd | 31.1 |
| | PI 177423 | 5.1 | ± | 5.1 | abcde | 4.6 | ± | 1.0 | cde | −9.8 |
| | PI 536803 | 7.3 | ± | 1.9 | bcde | 5.3 | ± | 1.6 | def | −27 |
| | PI 536822 | 5.1 | ± | 2.7 | bcdef | 3.5 | ± | 3.0 | cdef | −31 |
| | PI 593426 | 8.4 | ± | 1.8 | abcde | 7.1 | ± | 1.5 | abcd | −15 |
| | PI 600773 | 7.8 | ± | 1.7 | abcd | 7.1 | ± | 2.0 | abcde | −9 |
| | PI 635075 | 7.1 | ± | 2.4 | abcde | 7 | ± | 0.5 | bcde | −1.4 |
| | PI 635077 | 7.4 | ± | 0.8 | bcd | 7.9 | ± | 0.8 | abc | 6.76 |
| | Prizehead | 5.9 | ± | 0.6 | cd | 5.5 | ± | 2.0 | cd | −6.8 |

[1] Dry mass (in gram) on per plant basis. [2] Means followed by different letters are significantly different from each other (*p* < 0.05) as assessed by Tukey's test. [3] % indicates the percentage change in each variety due to change in treatment levels (Control vs. Salt), where negative sign (−) indicates a percentage decrease in DM.

Parris Island (CT lettuce) showed a significant increase in DM by 31% compared to control. Crispino, PI 635077, and PI 635075 showed no effect of salt treatment when

compared to control. On the contrary, there was a significant decrease in the DM% (7% to 31.4%) when treated with salt compared to the control in PI 177423, Prizehead, PI 600773, PI 536803, Iceberg, and PI 536822. PI 536822 was found the most sensitive to salt treatment (31.4% decrease) in terms of DM among all CT lettuce. PI 171676 and PI212099 were two LT lettuce with a significant increase in DM (57% and 63%, respectively) under SS compared to control (Table 6). On the other hand, PI 171675 showed a significant reduction (77.2%) in the DM under salt stress compared to control, followed by a decrease in DM by 15.7% to 36.6% in PI 601488, Ruby, PI 358018, and PI 601339. Burgundy Delight is the only RT lettuce with a significant increase in DM by 30% when subjected to SS compared to control. PI 536708 and PI 612664 were unaffected by salt treatment. However, the other genotypes (PI 274366, PI 289023, PI 613577, PI 278066, and Green Forest) showed a significant reduction in DM ranging from 10% to up to 30.7%.

**Table 6.** Dry mass (DM) (mean $\pm$ standard deviation, *n* = 6) of leaf-and romaine-type lettuce cultivars grown under control and salt-stress conditions (100 mM) after 42 days of sowing (10 days after salt treatment).

| Type | Variety | DM [1] | | | | | | | | %[3] |
|------|---------|--------|---|---|---|---|---|---|---|------|
| | | Control | | | | Salt | | | | |
| Leaf | PI 171675 | 7.9 | $\pm$ | 3 | abc[2] | 1.8 | $\pm$ | 0.5 | f | −77 |
| | PI 171676 | 1.9 | $\pm$ | 2 | f | 3 | $\pm$ | 4 | bcde | 57.9 |
| | PI 175737 | 3.2 | $\pm$ | 1.5 | ef | 3.4 | $\pm$ | 0.1 | ef | 6.25 |
| | PI 212099 | 3 | $\pm$ | 2.8 | cdef | 4.9 | $\pm$ | 1.9 | bcde | 63.3 |
| | PI 358018 | 5.8 | $\pm$ | 3.6 | abcde | 4.6 | $\pm$ | 0.9 | cde | −21 |
| | PI 601339 | 5.7 | $\pm$ | 1.5 | cdef | 4.8 | $\pm$ | 1.4 | cd | −16 |
| | PI 601488 | 8.3 | $\pm$ | 2.8 | abcde | 5.3 | $\pm$ | 0.4 | cd | −36 |
| | Ruby | 7.6 | $\pm$ | 1.8 | abcd | 5.5 | $\pm$ | 2.7 | abcd | −28 |
| Romaine | Burgundy Delight | 4.3 | $\pm$ | 3.2 | cdef | 5.6 | $\pm$ | 2.6 | bcde | 30.2 |
| | Green Forest | 10.1 | $\pm$ | 3.8 | a | 7 | $\pm$ | 2.4 | abc | −31 |
| | PI 278066 | 7.4 | $\pm$ | 2.5 | abcd | 5.7 | $\pm$ | 1.1 | cde | −23 |
| | PI 289023 | 5.7 | $\pm$ | 0.7 | bcde | 4.9 | $\pm$ | 1.0 | cde | −14 |
| | PI 536708 | 8.6 | $\pm$ | 2.6 | abcd | 8.2 | $\pm$ | | abc | −4.7 |
| | PI 612664 | 3.6 | $\pm$ | 1.2 | de | 3.7 | $\pm$ | 3.5 | cdef | 2.78 |
| | PI 613577 | 5.1 | $\pm$ | 1.5 | cde | 4 | $\pm$ | 1.7 | def | −22 |
| | PI 274366 | 4.0 | $\pm$ | 1.8 | def | 3.6 | $\pm$ | 0.7 | def | −10 |

[1] Dry mass (in gram) on per plant basis. [2] Means followed by different letters are significantly different from each other (*p* < 0.05) as assessed by Tukey's test. [3] % indicates the percentage change in each variety due to change in treatment levels (Control vs. Salt), where negative sign (−) indicates a percentage decrease in DM.

*2.2. Physiological Traits*

2.2.1. Chlorophyll Index

The effect of SS on the chlorophyll index (CI) of BH-type lettuce cultivars is shown in Table 7. The CI content of four BH-type lettuce, namely, PI 634671, Buttercrunch-1, PI 342448, and PI 615052, increased significantly by 21.6%, 19.4%, 19%, and 9.3%, respectively, under the SS compared to control. However, the other eight BH-type lettuce did not thrive well under the SS compared to control. Likewise, four CH-type lettuce (PI 593426, PI 536803, PI 635077, and Parris Island) showed better performance under SS compared to the control with an increase in CI content by up to 44% (Table 7). Similarly, Crispino and PI 600773 were moderately salt tolerant. However, there was a significant decrease in CI content in PI 536822, PI 177423, Prizehead, PI 635075, and Iceberg compared to control, as shown in Table 7.

The CI content in six out of eight LT lettuce cultivars was significantly reduced under SS compared to control (Table 8). Ruby and PI 601339 were affected the most by the SS (decrease by 30–32%). However, PI 175737 and PI 212099 thrived well under the SS compared to control.

**Table 7.** Chlorophyll index (CI) (mean $\pm$ standard deviation, $n = 6$) of butterhead- and crisphead-type lettuce cultivars grown under control and salt-stress conditions (100 mM) after 42 days of sowing (10 days after salt treatment).

| Type | Variety | CI [1] | | | | | | | | % [3] |
|------|---------|--------|---|---|---|---|---|---|---|------|
| | | Control | | | | Salt | | | | |
| Butterhead | Burpee Bibb | 14 | $\pm$ | 5.1 | abcd [2] | 11.3 | $\pm$ | 2.8 | e | −18 |
| | Buttercrunch-1 | 9.8 | $\pm$ | 2.5 | e | 11.7 | $\pm$ | 1.1 | e | 19.4 |
| | Buttercrunch-2 | 13 | $\pm$ | 1.5 | de | 10.9 | $\pm$ | 1.4 | e | −17 |
| | Hybrid Bibb | 14 | $\pm$ | 4.4 | bcde | 12.4 | $\pm$ | 3.7 | de | −8.1 |
| | PI 273606 | 7 | $\pm$ | 3.4 | efgh | 5.6 | $\pm$ | 1.8 | fghi | −20 |
| | PI 342448 | 6.3 | $\pm$ | 2.1 | fghi | 7.5 | $\pm$ | 1.4 | fghi | 19 |
| | PI 342515 | 6.6 | $\pm$ | 1.2 | fghi | 6.5 | $\pm$ | 1.2 | fghi | −1.5 |
| | PI 358020 | 5.8 | $\pm$ | 2.7 | ghi | 4.1 | $\pm$ | 3.1 | i | −29 |
| | PI 372908 | 8.2 | $\pm$ | 3.7 | e | 7 | $\pm$ | 0.9 | efgh | −15 |
| | PI 615052 | 7.5 | $\pm$ | 1.2 | efgh | 8.2 | $\pm$ | 3.5 | efgh | 9.33 |
| | PI 634671 | 3.7 | $\pm$ | 3.3 | i | 4.5 | $\pm$ | 3.6 | ghi | 21.6 |
| Crisphead | Crispino | 7.3 | $\pm$ | 1.1 | efg | 7.3 | $\pm$ | 1.9 | fghi | 0 |
| | Iceberg | 7.2 | $\pm$ | 2.1 | efgh | 6.5 | $\pm$ | 5.3 | fghi | −9.7 |
| | Parris Island | 13 | $\pm$ | 3 | 6 | 14.3 | $\pm$ | 2.9 | abcd | 14.4 |
| | PI 177423 | 9.5 | $\pm$ | 1.6 | e | 7.4 | $\pm$ | 3.5 | ef | −22 |
| | PI 536803 | 6.6 | $\pm$ | 2.7 | efgh | 8.3 | $\pm$ | 1.4 | e | 25.8 |
| | PI 536822 | 5.6 | $\pm$ | 2.3 | fghi | 3.7 | $\pm$ | 3.2 | i | −34 |
| | PI 593426 | 5.2 | $\pm$ | 1.5 | ghi | 7.5 | $\pm$ | 1.8 | efgh | 44.2 |
| | PI 600773 | 15 | $\pm$ | 1.1 | abcd | 14.4 | $\pm$ | 0.6 | abcd | −0.7 |
| | PI 635075 | 14 | $\pm$ | 3 | abcd | 11.4 | $\pm$ | 3 | de | −16 |
| | PI 635077 | 11 | $\pm$ | 4.1 | e | 13.8 | $\pm$ | 3.7 | bcde | 25.5 |
| | Prizehead | 6.8 | $\pm$ | 3.2 | efgh | 5.4 | $\pm$ | 1 | fghi | −21 |

[1] Chlorophyll index on per plant basis. [2] Means followed by different letters are significantly different from each other ($p < 0.05$) as assessed by Tukey's test. [3] % indicates the percentage change in each variety due to change in treatment levels (Control vs. Salt), where negative sign (−) indicates a percentage decrease in CI.

**Table 8.** Chlorophyll index (CI) (mean $\pm$ standard deviation, $n = 6$) of leaf- and romaine-type lettuce cultivars grown under control and salt-stress conditions (100 mM) after 42 days of sowing (10 days after salt treatment).

| Type | Variety | CI [1] | | | | | | | | % [3] |
|------|---------|--------|---|---|---|---|---|---|---|------|
| | | Control | | | | Salt | | | | |
| Leaf | PI 171675 | 5.5 | $\pm$ | 4.1 | fghi [2] | 4 | $\pm$ | 2.5 | i | −27 |
| | PI 171676 | 7.1 | $\pm$ | 5.3 | efgh | 5.4 | $\pm$ | 1 | ghi | −24 |
| | PI 175737 | 4.2 | $\pm$ | 5.3 | ghi | 4.8 | $\pm$ | 4.3 | ghi | 14.3 |
| | PI 212099 | 4.1 | $\pm$ | 2.1 | i | 4.9 | $\pm$ | 7.2 | ghi | 19.5 |
| | PI 358018 | 7.1 | $\pm$ | 0.6 | fghi | 6.4 | $\pm$ | 2.4 | fghi | −9.9 |
| | PI 601339 | 9.5 | $\pm$ | 3 | e | 6.6 | $\pm$ | 2.4 | efgh | −31 |
| | PI 601488 | 11.4 | $\pm$ | 2.7 | e | 9.6 | $\pm$ | 2.6 | e | −16 |
| | Ruby | 6.4 | $\pm$ | 2.2 | fghi | 4.3 | $\pm$ | 0.9 | hi | −33 |
| Romaine | Burgundy Delight | 13.4 | $\pm$ | 1.5 | abcd | 15.3 | $\pm$ | 1.4 | ab | 14.2 |
| | Green Forest | 15.4 | $\pm$ | 2.2 | a | 11.3 | $\pm$ | 1.6 | e | −27 |
| | PI 278066 | 8.5 | $\pm$ | 1.4 | e | 5.8 | $\pm$ | 2.6 | efgh | −32 |
| | PI 289023 | 8 | $\pm$ | 3.7 | e | 5.6 | $\pm$ | 3.1 | efgh | −30 |
| | PI 536708 | 14.9 | $\pm$ | 0.8 | abc | 13.6 | $\pm$ | 4.7 | abcd | −8.7 |
| | PI 612664 | 9 | $\pm$ | 4.7 | e | 11.5 | $\pm$ | 4.6 | e | 27.8 |
| | PI 613577 | 9.6 | $\pm$ | 3.9 | e | 9.4 | $\pm$ | 0.5 | e | −2.1 |
| | PI 274366 | 4.1 | $\pm$ | 1.2 | i | 2.6 | $\pm$ | 1.8 | i | −37 |

[1] Chlorophyll index on per plant basis. [2] Means followed by different letters are significantly different from each other ($p < 0.05$) as assessed by Tukey's test. [3] % indicates the percentage change in each variety due to change in treatment levels (Control vs. Salt), where negative sign (−) indicates a percentage decrease in CI.

Only two romaine-type lettuce cultivars showed an increase in CI content under SS, i.e., Burgundy Delight (14%) and PI 612664 (27.8%). CI content in the other six RT lettuce

cultivars was significantly decreased from 8% to 36.5% under SS compared to control (Table 8).

### 2.2.2. Flavonoid Index

The effect of SS on flavonoid index (FI) in different BH-type lettuce cultivars is shown in Table 9. The result demonstrated that Burpee Bibb and PI 358020 remained unaffected by salt application compared to control. Furthermore, PI 615052 and PI 342448 showed an increase in FI under SS compared to control. On the contrary, Buttercrunch-1,2, PI 634671, PI 372908, PI 342515, and PI 273606 showed decreased FI under SS compared to control. Out of 11 CH-type lettuce cultivars, six cultivars showed no effect of SS in FI level (Table 9). However, FI of Crispino and PI 635077 were reduced significantly under SS by 20% and 10% compared to control. On the other hand, there was a significant increase in FI level in PI 177423 and PI 536822 under SS by 10% compared to control.

**Table 9.** Flavonoid index (FI) (mean ± standard deviation, *n* = 6) of butterhead- and crisphead-type lettuce cultivars grown under control and salt-stress conditions (100 mM) after 42 days of sowing (10 days after salt treatment).

| Type | Variety | FI [1] | | | | | | | | %[3] |
|---|---|---|---|---|---|---|---|---|---|---|
| | | Control | | | | Salt | | | | |
| Butterhead | Burpee Bibb | 0.9 | ± | 0.1 | fg[2] | 0.9 | ± | 0.3 | fg | 0 |
| | Buttercrunch-1 | 1.3 | ± | 0.1 | abcd | 1.2 | ± | 0.2 | abcde | −7.7 |
| | Buttercrunch-2 | 1.3 | ± | 0.2 | abcd | 1.2 | ± | 0 | abcde | −7.7 |
| | Hybrid Bibb | 1.6 | ± | 0.2 | abcd | 1.5 | ± | 0.1 | a | −6.3 |
| | PI 273606 | 1.1 | ± | 0.3 | defg | 1 | ± | 0.2 | efg | −9.1 |
| | PI 342448 | 1.1 | ± | 0.2 | cdef | 1.3 | ± | 0.3 | abcde | 18.2 |
| | PI 342515 | 1.2 | ± | 0.1 | abcde | 1.1 | ± | 0 | cdef | −8.3 |
| | PI 358020 | 1.3 | ± | 0.1 | abcde | 1.3 | ± | 0.3 | abcde | 0 |
| | PI 372908 | 1.2 | ± | 0.1 | abcde | 1.1 | ± | 0.2 | defg | −8.3 |
| | PI 615052 | 1.1 | ± | 0.1 | defg | 1.2 | ± | 0.4 | abcde | 9.09 |
| | PI 634671 | 1.2 | ± | 0.2 | defg | 1.1 | ± | 0.2 | efg | −8.3 |
| Crisphead | Crispino | 1 | ± | 0.1 | g | 0.8 | ± | 0.1 | g | −20 |
| | Iceberg | 1.4 | ± | 0.2 | abc | 1.4 | ± | 0 | abc | 0 |
| | Parris Island | 1.5 | ± | 0 | a | 1.4 | ± | 0.1 | a | −6.7 |
| | PI 177423 | 1 | ± | 0.2 | efg | 1.1 | ± | 0.1 | cdef | 10 |
| | PI 536803 | 1 | ± | 0.1 | efg | 1 | ± | 0 | fg | 0 |
| | PI 536822 | 1.1 | ± | 0.1 | efg | 1.2 | ± | 0.2 | abcde | 9.09 |
| | PI 593426 | 0.8 | ± | 0.1 | g | 0.8 | ± | 0.1 | g | 0 |
| | PI 600773 | 1 | ± | 0.2 | fg | 1 | ± | 0.1 | efg | 0 |
| | PI 635075 | 1 | ± | 0.1 | fg | 1 | ± | 0.2 | efg | 0 |
| | PI 635077 | 1 | ± | 0.4 | fg | 0.9 | ± | 0.2 | fg | −10 |
| | Prizehead | 1.4 | ± | 0.1 | abc | 1.4 | ± | 0.2 | abc | 0 |

[1] Flavonoid index on per plant basis. [2] Means followed by different letters are significantly different from each other (*p* < 0.05) as assessed by Tukey's test. [3] % indicates the percentage change in each variety due to change in treatment levels (Control vs. Salt), where negative sign (−) indicates a percentage decrease in FI.

Four LT lettuce cultivars, PI 601488, Ruby, PI 171675, and PI 358018, thrived well under SS with an increase in FI by 8.3–25% compared to control (Table 10). However, the FI of the other four LT lettuce cultivars was significantly reduced by 7% to 21.5% under SS compared to control. PI 612664, a romaine-type lettuce cultivar, was reported unaffected under the SS compared to control. On the other hand, four RT lettuce cultivars, Green Forest, PI 289023, PI 274366, and PI 278066, showed an increasing trend in FI when treated with salt compared to control. In contrast, three other RT lettuce cultivars (Burgundy Delight, PI 536708, and PI 613577) showed a significant decrease in FI by 6% to 8% compared to control (Table 10).

**Table 10.** Flavonoid index (FI) (mean $\pm$ standard deviation, *n* = 6) of leaf- and romaine-type lettuce cultivars grown under control and salt-stress conditions (100 mM) after 42 days of sowing (10 days after salt treatment).

| Type | Variety | FI [1] | | | | | | | | % [3] |
|---|---|---|---|---|---|---|---|---|---|---|
| | | Control | | | | Salt | | | | |
| Leaf | PI 171675 | 1 | $\pm$ | 0.2 | abcde [2] | 1.1 | $\pm$ | 0.2 | cdef | 10 |
| | PI 171676 | 1.4 | $\pm$ | 0.2 | abc | 1.1 | $\pm$ | 0 | bcdef | −21 |
| | PI 175737 | 1.2 | $\pm$ | 0.1 | abcde | 1 | $\pm$ | 0.1 | efg | −17 |
| | PI 212099 | 1.1 | $\pm$ | 0.2 | efg | 1 | $\pm$ | 0.2 | efg | −9.1 |
| | PI 358018 | 1.2 | $\pm$ | 0.2 | bcdef | 1.3 | $\pm$ | 0.2 | abcd | 8.33 |
| | PI 601339 | 1.4 | $\pm$ | 0.4 | a | 1.3 | $\pm$ | 0.3 | abcd | −7.1 |
| | PI 601488 | 0.8 | $\pm$ | 0.1 | g | 1 | $\pm$ | 0.2 | fg | 25 |
| | Ruby | 1.1 | $\pm$ | 0.2 | cdefg | 1.3 | $\pm$ | 0.1 | abcde | 18.2 |
| Romaine | Burgundy Delight | 1.6 | $\pm$ | 0.1 | a | 1.5 | $\pm$ | 0.2 | a | −6.3 |
| | Green Forest | 1.3 | $\pm$ | 0.1 | abc | 1.4 | $\pm$ | 0 | ab | 7.69 |
| | PI 278066 | 1.4 | $\pm$ | 0.3 | abc | 1.9 | $\pm$ | 0.1 | a | 35.7 |
| | PI 289023 | 1.2 | $\pm$ | 0.2 | cdefg | 1.3 | $\pm$ | 0.8 | abcde | 8.33 |
| | PI 536708 | 1.3 | $\pm$ | 0.1 | abcde | 1.2 | $\pm$ | 0.2 | abcde | −7.7 |
| | PI 612664 | 1.2 | $\pm$ | 0.2 | abcde | 1.2 | $\pm$ | 0.2 | abcde | 0 |
| | PI 613577 | 1.5 | $\pm$ | 0.3 | a | 1.4 | $\pm$ | 0.2 | a | −6.7 |
| | PI 274366 | 1.1 | $\pm$ | 0.2 | abc | 1.4 | $\pm$ | 0.2 | a | 27.3 |

[1] Flavonoid index on per plant basis. [2] Means followed by different letters are significantly different from each other (*p* < 0.05) as assessed by Tukey's test. [3] % indicates the percentage change in each variety due to change in treatment levels (Control vs. Salt), where negative sign (−) indicates a percentage decrease in FI.

### 2.2.3. Nitrogen Balance Index

Nitrogen balance index (NB) was significantly affected in 5 out of 11 BH lettuces, where Buttercrunch-1 and PI 634671 were reported to significantly increase NB by 48.6% and 36.4%, respectively (Table 11), under SS compared to control. On the other hand, salt treatment did not show a significant change in NB in PI 273606 and PI 342448 compared to control. Besides, PI 358020 and PI 615052 showed a significant reduction in NB by 38% and 28% under SS compared to control. PI 536803 and PI 593426 are CH lettuce which responded significantly higher in terms of NB when treated with salt as NB increased by 46% and 40%, respectively, compared to control. On the other hand, the NB index of PI 600773 and Crispino remained unaffected by salt treatment, as shown in Table 11. Besides, PI 536822 and PI 635075 showed a significant reduction in NB level by 29% and 36.9%, respectively, under SS when compared to control.

There was a significant increase in NB of three LT lettuce (PI 212099, PI 175737, and PI 171676) by 56.4%, 37.5%, and 22.5% when treated with salt as compared to control (Table 12). On the other hand, compared to control, NB decreased significantly by 19% to 40% in PI 358018, PI 601488, and Ruby under salt treatment. Besides, the NB in RT lettuce (Burgundy Delight and PI 612664) was significantly increased by 16% and 37.5% due to salt treatment compared to control. There was a significant reduction in NB in PI 278066 (44.4%), PI 289023 (31.5%), and Green Forest (23.4%) when treated with salt as compared to control.

**Table 11.** Nitrogen balance index (NB (mean ± standard deviation, *n* = 4) of butterhead- and crisphead-type lettuce cultivars grown under control and salt-stress conditions (100 mM) after 42 days of sowing (10 days after salt treatment).

| Type | Variety | NB [1] | | | | | | | % [3] |
|------|---------|---------|---|---|---|---|---|---|---|
| | | Control | | | | Salt | | | |
| Butterhead | Burpee Bibb | 15 | ± | 8.6 | ab [2] | 12.8 | ± | 5.6 | bc | −16 |
| | Buttercrunch-1 | 7.2 | ± | 1.8 | de | 10.7 | ± | 3.2 | bcde | 48.6 |
| | Buttercrunch-2 | 11 | ± | 2.1 | cde | 9.7 | ± | 1.1 | cde | −12 |
| | Hybrid Bibb | 7.6 | ± | 5.4 | de | 9.1 | ± | 4 | de | 19.7 |
| | PI 273606 | 6.8 | ± | 10 | de | 6.5 | ± | 3.8 | de | −4.4 |
| | PI 342448 | 5.3 | ± | 1.4 | de | 5.3 | ± | 3.6 | de | 0 |
| | PI 342515 | 6.3 | ± | 0.3 | de | 7.2 | ± | 0.8 | de | 14.3 |
| | PI 358020 | 5.2 | ± | 1.4 | de | 3.2 | ± | 4.5 | e | −38 |
| | PI 372908 | 7.2 | ± | 3.6 | de | 8.9 | ± | 3.9 | de | 23.6 |
| | PI 615052 | 9.7 | ± | 1.7 | de | 6.9 | ± | 8.3 | de | −29 |
| | PI 634671 | 3.3 | ± | 5.5 | e | 4.5 | ± | 5.4 | e | 36.4 |
| Crisphead | Crispino | 7.9 | ± | 1.3 | cde | 8.4 | ± | 1.2 | de | 6.33 |
| | Iceberg | 5.2 | ± | 5 | e | 4.6 | ± | 6.3 | e | −12 |
| | Parris Island | 9.6 | ± | 1.7 | de | 10.9 | ± | 1.4 | bcde | 13.5 |
| | PI 177423 | 8.6 | ± | 1.6 | de | 6.6 | ± | 4.7 | de | −23 |
| | PI 536803 | 6.5 | ± | 1.9 | de | 9.5 | ± | 2.5 | de | 46.2 |
| | PI 536822 | 4.8 | ± | 2.7 | de | 3.4 | ± | 7.7 | e | −29 |
| | PI 593426 | 6.5 | ± | 2.7 | de | 9.1 | ± | 2.4 | de | 40 |
| | PI 600773 | 15 | ± | 0.2 | ab | 15.8 | ± | 1.0 | ab | 3.27 |
| | PI 635075 | 18 | ± | 2.6 | a | 11.3 | ± | 2.7 | bcd | −37 |
| | PI 635077 | 13 | ± | 11 | bc | 13.7 | ± | 3.7 | b | 7.03 |
| | Prizehead | 4.9 | ± | 3.7 | e | 4.1 | ± | 2.7 | e | −16 |

[1] Nitrogen balance index on per plant basis. [2] Means followed by different letters are significantly different from each other (*p* < 0.05) as assessed by Tukey's test. [3] % indicates the percentage change in each variety due to change in treatment levels (Control vs. Salt), where negative sign (−) indicates a percentage decrease in NB.

**Table 12.** Nitrogen balance index (NB (mean ± standard deviation, *n* = 4) of leaf- and romaine-type lettuce cultivars grown under control and salt-stress conditions (100 mM) after 42 days of sowing (10 days after salt treatment).

| Type | Variety | NB [1] | | | | | | | % [3] |
|------|---------|---------|---|---|---|---|---|---|---|
| | | Control | | | | Salt | | | |
| Leaf | PI 171675 | 6 | ± | 4.8 | de [2] | 5.9 | ± | 3.8 | de | −1.7 |
| | PI 171676 | 4.9 | ± | 5.5 | e | 6 | ± | 1.2 | de | 22.4 |
| | PI 175737 | 4 | ± | 4.2 | e | 5.5 | ± | 2.8 | de | 37.5 |
| | PI 212099 | 3.9 | ± | 4.8 | e | 6.1 | ± | 8.5 | de | 56.4 |
| | PI 358018 | 6.9 | ± | 1.3 | de | 5.6 | ± | 3 | de | −19 |
| | PI 601339 | 6.7 | ± | 4.1 | de | 5.9 | ± | 4.6 | de | −12 |
| | PI 601488 | 13.9 | ± | 2.9 | b | 10 | ± | 6 | cde | −28 |
| | Ruby | 5.9 | ± | 2.1 | de | 3.6 | ± | 1.9 | e | −39 |
| Romaine | Burgundy Delight | 8.1 | ± | 1.8 | de | 9.4 | ± | 1.8 | de | 16 |
| | Green Forest | 12.4 | ± | 1.4 | bcd | 9.5 | ± | 0.8 | de | −23 |
| | PI 278066 | 7.2 | ± | 3.4 | de | 4 | ± | 2.3 | e | −44 |
| | PI 289023 | 7.3 | ± | 4.4 | de | 5 | ± | 2.3 | de | −32 |
| | PI 536708 | 12.5 | ± | 3.3 | bcd | 11.6 | ± | 6.4 | bcd | −7.2 |
| | PI 612664 | 8 | ± | 6.2 | de | 11 | ± | 5.4 | bcde | 37.5 |
| | PI 613577 | 6.5 | ± | 3.8 | de | 6.3 | ± | 0.7 | de | −3.1 |
| | PI 274366 | 3.5 | ± | 1.9 | e | 2.9 | ± | 2.5 | e | −17 |

[1] Nitrogen balance index on per plant basis. [2] Means followed by different letters are significantly different from each other (*p* < 0.05) as assessed by Tukey's test. [3] % indicates the percentage change in each variety due to change in treatment levels (Control vs. Salt), where negative sign (−) indicates a percentage decrease in NB.

### 2.2.4. Anthocyanin Index

The response of anthocyanin index (AI) in BH-type lettuce cultivars under SS is shown in Table 13. The AI levels of Burpee Bibb and PI 342515 under SS were increased significantly by 100% and 50%, respectively, compared to control. On the other hand, AI levels of PI 372908 and PI 342448 were significantly reduced by 33.33% and 25%, respectively, under SS compared to control. AI level of 7 out of 11 BH-type lettuce cultivars remained unaffected under SS, as shown in Table 13. AI levels of two crisphead-type (PI 600773 and PI 635075) lettuce were significantly increased by 100% under SS compared to control. At the same time, the other four (Parris Island, PI 177423, Iceberg, and Prizehead) were significantly reduced up to 50%. The AI level of five CH-type cultivars (Crispino, PI 536803, PI 536822, and PI 635077) was unaffected under SS compared to control (Table 13).

**Table 13.** Anthocyanin Index (AI) (mean ± standard deviation, *n* = 6) of butterhead- and crisphead-type lettuce cultivars grown under control and salt-stress conditions (100 mM) after 42 days of sowing (10 days after salt treatment).

| Type | Variety | AI [1] | | | | | | | % [3] |
|---|---|---|---|---|---|---|---|---|---|
| | | Control | | | | Salt | | | |
| Butterhead | Burpee Bibb | 0.1 | ± | 0.1 | klm [2] | 0.2 | ± | 0.1 | jk | 100 |
| | Buttercrunch-1 | 0.2 | ± | 0.1 | lm | 0.2 | ± | 0 | m | 0 |
| | Buttercrunch-2 | 0.2 | ± | 0.2 | lm | 0.2 | ± | 0 | lm | 0 |
| | Hybrid Bibb | 0.2 | ± | 0 | m | 0.2 | ± | 0 | m | 0 |
| | PI 273606 | 0.2 | ± | 0 | klm | 0.2 | ± | 0 | klm | 0 |
| | PI 342448 | 0.4 | ± | 0 | hij | 0.3 | ± | 0.1 | j | −25 |
| | PI 342515 | 0.2 | ± | 0.2 | klm | 0.3 | ± | 0.2 | jk | 50 |
| | PI 358020 | 0.2 | ± | 0 | jk | 0.2 | ± | 0 | jk | 0 |
| | PI 372908 | 0.3 | ± | 0 | jk | 0.2 | ± | 0 | jk | −33.3 |
| | PI 615052 | 0.2 | ± | 0.1 | j | 0.2 | ± | 0.1 | j | 0 |
| | PI 634671 | 0.2 | ± | 0.1 | jkl | 0.2 | ± | 0.1 | klm | 0 |
| Crisphead | Crispino | 0.2 | ± | 0.1 | klm | 0.2 | ± | 0.1 | klm | 0 |
| | Iceberg | 0.4 | ± | 0 | ij | 0.3 | ± | 0.1 | j | −25 |
| | Parris Island | 0.2 | ± | 0.1 | klm | 0.1 | ± | 0.1 | m | −50 |
| | PI 177423 | 0.3 | ± | 0 | jk | 0.2 | ± | 0 | jk | −33 |
| | PI 536803 | 0.2 | ± | 0.1 | klm | 0.2 | ± | 0.1 | klm | 0 |
| | PI 536822 | 0.4 | ± | 0 | hi | 0.4 | ± | 0 | ij | 0 |
| | PI 593426 | 0.2 | ± | 0 | klm | 0.2 | ± | 0 | klm | 0 |
| | PI 600773 | 0.1 | ± | 0.1 | m | 0.2 | ± | 0 | m | 100 |
| | PI 635075 | 0.1 | ± | 0 | m | 0.2 | ± | 0.1 | lm | 100 |
| | PI 635077 | 0.2 | ± | 0.1 | m | 0.2 | ± | 0.1 | klm | 0 |
| | Prizehead | 0.6 | ± | 0 | e | 0.5 | ± | 0 | g | −17 |

[1] Anthocyanin index on per plant basis. [2] Means followed by different letters are significantly different from each other ($p < 0.05$) as assessed by Tukey's test. [3] % indicates the percentage change in each variety due to change in treatment levels (Control vs. Salt), where negative sign (−) indicates a percentage decrease in AI.

LT cultivars (Ruby, PI 171675, and PI 175737) showed a significant increase in AI level under SS by 20% to 50% compared to control, while PI 601339 showed a 20% decrease in AI level under SS (Table 14). Four leaf-type cultivars (PI 171676, PI 212099, PI 358018, and PI 601488) showed no effect of SS in the AI level. Green Forest (RT) was the only cultivar with an increased AI level by 100% under SS compared to control. PI 613577 and Burgundy Delight, the other two RT lettuce cultivars, showed decreased AI levels by 9% and 12%, respectively, under SS compared to control. The other five RT cultivars were unaffected by salt treatment (Table 14).

**Table 14.** Anthocyanin Index (AI) (mean ± standard deviation, *n* = 6) of leaf- and romaine-type lettuce cultivars grown under control and salt-stress conditions (100 mM) after 42 days of sowing (10 days after salt treatment).

| Type | Variety | AI [1] | | | | | | | | % [3] |
|---|---|---|---|---|---|---|---|---|---|---|
| | | Control | | | | Salt | | | | |
| Leaf | PI 171675 | 0.2 | ± | 0.1 | klm [2] | 0.3 | ± | 0 | jk | 50 |
| | PI 171676 | 0.2 | ± | 0.3 | klm | 0.2 | ± | 0 | klm | 0 |
| | PI 175737 | 0.2 | ± | 0 | klm | 0.3 | ± | 0 | jk | 50 |
| | PI 212099 | 0.2 | ± | 0 | klm | 0.2 | ± | 0.1 | jk | 0 |
| | PI 358018 | 0.2 | ± | 0 | klm | 0.2 | ± | 0 | jk | 0 |
| | PI 601339 | 0.5 | ± | 0 | fg | 0.4 | ± | 0 | hi | −20 |
| | PI 601488 | 0.2 | ± | 0 | m | 0.2 | ± | 0 | klm | 0 |
| | Ruby | 0.5 | ± | 0 | fg | 0.6 | ± | 0 | ef | 20 |
| Romaine | Burgundy Delight | 1.1 | ± | 0 | a | 1 | ± | 0 | b | −9.1 |
| | Green Forest | 0.1 | ± | 0.1 | m | 0.2 | ± | 0.1 | klm | 100 |
| | PI 278066 | 0.2 | ± | 0 | klm | 0.2 | ± | 0 | jk | 0 |
| | PI 289023 | 0.2 | ± | 0 | klm | 0.2 | ± | 0 | klm | 0 |
| | PI 536708 | 0.1 | ± | 0 | m | 0.1 | ± | 0 | m | 0 |
| | PI 612664 | 0.3 | ± | 0 | j | 0.3 | ± | 0 | jk | 0 |
| | PI 613577 | 0.8 | ± | 0.1 | c | 0.7 | ± | 0 | d | −13 |
| | PI 274366 | 0.3 | ± | 0 | jk | 0.3 | ± | 0 | jk | 0 |

[1] Anthocyanin index on per plant basis. [2] Means followed by different letters are significantly different from each other (*p* < 0.05) as assessed by Tukey's test. [3] % indicates the percentage change in each variety due to change in treatment levels (Control vs. Salt), where negative sign (−) indicates a percentage decrease in AI.

### 2.2.5. Salt Tolerance Coefficient Index

Salt tolerance coefficient index (STCI) values varied from 0.75 to 1.49, and four different salt-tolerant groups were classified based on STCI values and standard deviation (Table 15). PI 171675, PI 536822, Ruby, PI 601339, Green Forest, PI 601488, PI 613577, PI 278066, Hybrid Bibb, PI 273606, and Burpee Bibb were classified as highly salt-sensitive cultivars, with STCI values less than or equal to 0.89. PI 372908, PI 358018, PI 274366, Prizehead, PI 177423, PI 289023, Iceberg, PI 635075, PI 342515, PI 358020, PI 536708, Buttercrunch-2, Crispino, PI 600773, and PI 536803 were classified as low salt-tolerant cultivars, with STCI value ranging from 0.90 to 1.03. Lettuce genotypes PI 342448, PI 615052, PI 175737, PI 593426, PI 634671, PI 612664, Parris Island, Burgundy Delight, and PI 635077 were classified as moderately salt-tolerant cultivars, with STCI values ranging from 1.04 to 1.17. Moreover, PI 171676, Buttercrunch-1, and PI 212099 were classified as highly salt-tolerant cultivars, with STCI values above 1.18.

**Table 15.** Tolerance classification (TC) of lettuce cultivars into various salt-tolerance groups (highly salt-tolerant, moderately salt-tolerant, low salt-tolerant and highly salt-sensitive) based on the Total salt tolerance coefficient index (STCI). Each STCI value is the sum of individual salt tolerance coefficient index values of three morphological and four biological parameters.

| Tolerance Classification | Salt Tolerance Coefficient Index Value | Cultivars |
|---|---|---|
| Highly salt-sensitive | (≤0.89) | PI 171675, PI 536822, Ruby, PI 601339, Green Forest, PI 601488, PI 613577, PI 278066, Hybrid Bibb, PI 273606, Burpee Bibb |
| Low salt tolerance | (0.90–1.03) | PI 372908, PI 358018, PI 274366, Prizehead, PI 177423, PI 289023, Iceberg, PI 635075, PI 342515, PI 358020, PI 536708, Buttercrunch-2, Crispino, PI 600773, PI 536803 |
| Moderate salt tolerance | (1.04–1.17) | PI 342448, PI 615052, PI 175737, PI 593426, PI 634671, PI 612664, Parris Island, Burgundy Delight, PI 635077 |
| High salt tolerance | (≥1.18) | PI 171676, Buttercrunch-1, PI 212099 |

### 2.3. Principal Component Analysis

Principal Component Analysis (PCA) was performed for all morphological and physiological traits of 38 genotypes in response to SS based on STCI values (Figure 1). The sum of principal components PC1 and PC2 explained 64.8% of the variations among the 38 lettuce genotypes. CI, DM, FM, and NB had the highest positive loading value, i.e., ~0.5, and AI and FI had the lowest loading value, i.e., ~−0.2, indicating the highest influence on PC1 and PC2. Based on the loading plot value, leaf number had the lowest influence on PC1 and PC2 (Figure 2). Similarly, PCA results demonstrated that PI 212099, PI 171676, Parris Island, Buttercrunch-1, Burgundy Delight, and PI 612664 were highly salt-tolerant. On the contrary, PI 171675, PI 278066, Ruby, PI 601488, Burpee Bibb, and Hybrid Bibb were highly salt-sensitive, which coincides with the classification of salt-tolerant groups based on STCI values in Table 15.

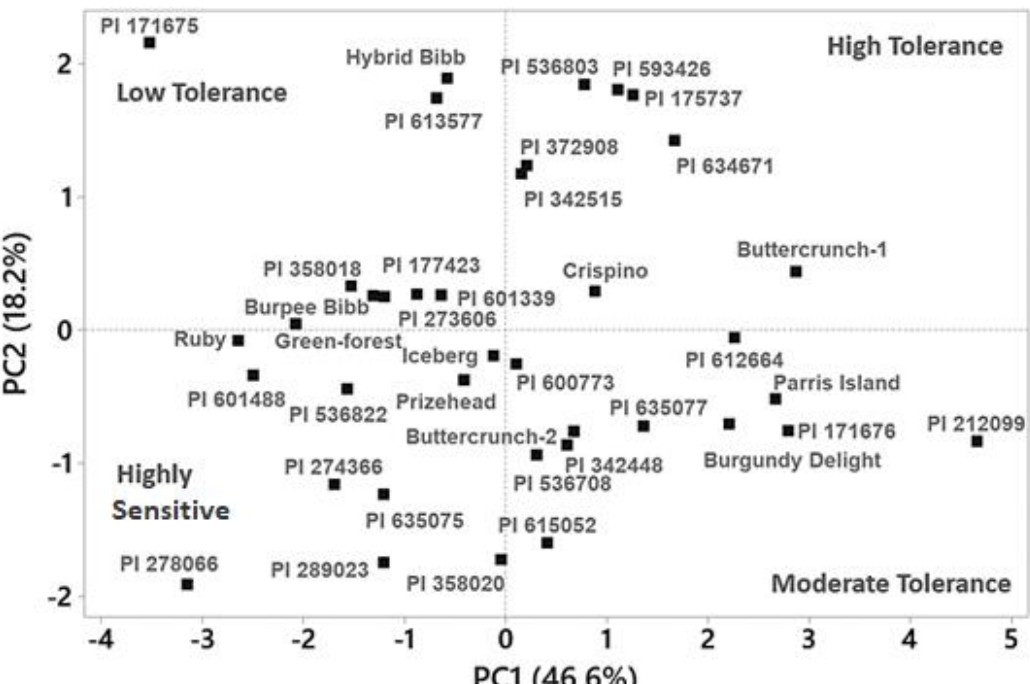

**Figure 1.** Principal component analysis (PCA) of first two principal components (PC) scores, PC1 and PC2 describing the classification of different lettuce genotypes into different salt-tolerant groups (high tolerance, moderate tolerance, low tolerance, and highly sensitive) based on seven morpho-physiological parameters measured 42 days after sowing for all the genotypes (10 days after salt treatment).

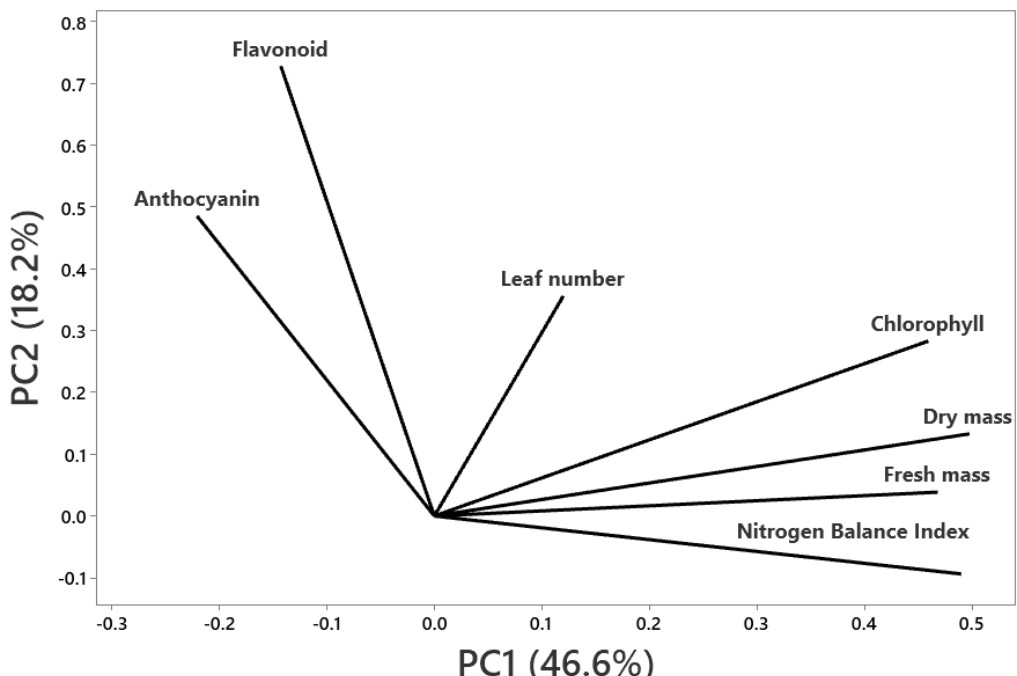

**Figure 2.** Loading plot in principal component analysis (PCA) of first two principal components (PC) scores, PC1 and PC2 describing the classification of different lettuce genotypes into different salt-tolerant groups (high tolerance, moderate tolerance, low tolerance, and highly sensitive) based on seven morpho-physiological parameters measured 42 days after sowing for all the genotypes.

## 3. Discussion

### 3.1. Response of Morphological Traits under Salt Stress

Identifying salt-tolerance traits in different types of lettuce is essential for successful commercial lettuce production in the soil as well as in hydroponic systems. The inhibitory effect of SS on the growth and development of lettuce and other leafy vegetables has been reported earlier by different authors [25–29]. This effect can be due to decreased accessible water necessary for crop growth, which causes an osmotic disturbance and, ultimately, impaired cell division [30–32]. Additionally, it is also widely understood that lettuce is moderately sensitive to salt and the range of sensitivity to SS varies from one genotype to another [20,33,34]. Our results demonstrated that 100 mM of NaCl concentration significantly affected LN, FM, DM, and physiological parameters such as NB, CI, FI, and AI. However, SS differs from one type of lettuce to another within the same type of lettuce cultivars as reported in previous studies [2]. For instance, PI 612664 and PI 635077 were highly salt-tolerant RT and BH lettuce cultivars based on LN. At the same time, PI 613577 and Hybrid Bibb have reported highly salt-sensitive RT and BH lettuce cultivars, respectively. The results from our study suggest that morphological and physiological traits of different lettuce cultivars towards SS are genotype-dependent, which is supported by earlier studies [2,35].

SS significantly affected LN compared to control conditions (Tables 1 and 2). The highest reduction in LN was recorded in PI 171675 (29.8%; LT) followed by PI 613577 (26.5%; RT), Hybrid Bibb (20.6%; BH), and PI 593426 (15.4%; CH). A previous study on ten-day-old *Vicia faba* (L.) bean plants exhibited fewer leaves when exposed to SS [36]. Similarly, increasing SS from 2.5 to 8.5 dS/m decreased LN by up to 36% in strawberries [37]. The reduction in leaf number in leaf-type lettuce cultivars by 6.6% due to SS (20 mM) was also reported earlier in lettuce [38]. In our study, the reduction in LN was more significant as the salt concentration used was five times higher than the salt concentration used in the study by Moncada et al. [38]. On the other hand, the highest increase in LN was recorded in PI 612664 (15.5%; RT) followed by PI 536708 (10.9%; RT), PI 635077 (9.3%; CH), PI 342448; BH), and PI 212099 (1.1%; LT). PI 342448 was also reported as moderately salt-tolerant by

Xu and Mou [2]. However, they reported PI 212099 as a salt-sensitive lettuce genotype, which is inconsistent with our result. In our study, PI 212099 was found unaffected in terms of LN and AI when subjected to SS. Besides, FM, DM, NB, and CI were increased when the plants were under SS, which suggested that PI 212099 thrives well under the SS condition.

SS also significantly impacted FM compared to control. The highest percentage loss in FM was recorded in PI 171675 (71.8%; LT) followed by PI 601488 (38.9%; LT), Ruby (35%; LT), Green Forest (29.7%; RT), PI 536822 (29.5%; CH), Hybrid bibb (24.3%; BH), and Burpee Bibb (7.6%; BH). In the previous study, Burpee Bibb and PI 17675 were identified as salt-sensitive [2,39]. However, Xu and Mou [2] reported Green Forest to be salt-tolerant in that study, which is recorded salt-sensitive based on FM in our research. On the other hand, PI 212099 (196%; LT), PI (104%; LT), Parris Island (66.9%; CH), Buttercrunch-1,2 (40–60%; BH), and Burgundy Delight (48%; RT) were highly salt-tolerant cultivars based on their absolute or relative growth. For example, Parris Island was reported as a salt-tolerant cultivar in a study by Shannon and McCreight [39]. Parris Island was also reported to have the lowest FM and DM loss in the study by Adhikari et al. [40]. Similarly, Prizehead (6.5%; CH) was also recorded as moderately salt-tolerant in this study, consistent with the study on lettuce by Xu and Mou [2].

SS significantly impacted shoot DM, as shown in Tables 5 and 6. DM of LT lettuce was significantly affected compared to others. The most salt-sensitive cultivars were PI 171675 (77.2%; LT) followed by PI 601488 (36%; LT), Burpee, and Hybrid Bibb (30–4.7%; BH), PI 536822 (31.3%; CH), and Green Forest (30.6%; RT). A previous study reported that RT cultivar generally showed decreased DM by 40% under 100 mM of salt concentration. Similarly, another study on LT cultivars reported almost 60% DM loss under 100 mM NaCl concentration compared to control [29]. However, two LT cultivars (PI 171676 and PI 212099) were found highly salt tolerant. The cultivars with high growth potential were highly salt-sensitive based on the percentage of growth decrease, whereas cultivars with high salt tolerance frequently had poor growth potential under control conditions [2].

### 3.2. Response of Physiological Traits under Salt Stress

Chlorophyll (CI) is an indicator of greenness in leafy vegetables such as lettuce. There is a series of progression or degradation of CI based on the stress they are subjected to [41]. CI degradation usually occurs when subjected to SS [35]. Our study demonstrated that PI 274366 (RT), PI 536822 (CH), Ruby (LT), and PI 171675 (BH) were significantly reduced by 30–37% under SS. These results were consistent with the reports presented earlier describing the inhibition of CI synthesis and activation of CI degradation due to SS [35,42,43]. Besides, increased severity of salinity on the CI in lettuce was also reported by Adhikari et al. [40]. On the contrary, PI 593426, PI 612664, PI 536803, PI634671, and PI 212099 showed poor accumulation of CI under control conditions and did well under the 100 mM salt solution. Broadly, this increase in chlorophyll content with increasing NaCl concentration was most likely caused by a decrease in chlorophyllase activity, which in turn promotes chlorophyll production and minimizes other detrimental consequences on membrane stability [44,45]. Besides, changes in cell anatomy occur in the plant when subjected to SS, resulting in smaller leaves and a higher chloroplast density per unit leaf area [46,47]. This could be one of the reasons behind higher CI and lower FM and DM in PI 593426 and PI 536803 in our study. Furthermore, under high salinity, lettuce may have changes in chlorophyll formation pathways that upregulate chlorophyll production or downregulate its breakdown [46,48,49]. This suggests that CI can increase or decrease under SS based on the chlorophyll pathway followed on that genotype. It has been reported that, under any environmental stress, phenolic compounds (including flavonols) accumulate, and there is a negative correlation between plant biomass production and any phenolic compounds [50]. For example, in our study, it was observed that lettuce (PI 171676), with the second-highest increase in FM (104%) under SS, had the lowest FI. Similarly, PI 278066, with the highest increase in FI (35.8%) under SS, showed a significant reduction (33%) in FM. Epidermal phenolic compounds produced under SS can function as

the obstacle that minimizes the chlorophyll excitation and, thus, affect the plant's overall biomass [51]. On the other hand, PI 342448 (BH) showed increased FM (13.5%) even under increased FI when treated with salt. It can be assumed that PI 342448 could withstand the SS by increasing flavonols accumulation in their leaves. When subjected to SS, there was a significant decrease in FI content in some cultivars such as Parris Island (50%) PI 372908 (33%), PI 342448 (25%), Iceberg (25%), and PI 601339 (20%). Our results were consistent with studies on lettuce [35] and pepper [52]. However, our study also demonstrated an increase in FI content even under SS in some cultivars such as PI 278066 (35.7%), PI 274366 (27.3%), PI 601488 (25%), and Ruby (18%). Our results are further supported by a study on multi-leaf lettuce by Garrido et al. [34] suggesting that the effect of NaCl is strongly influenced by plant genotype.

Besides, there is an inverse relation between flavonol and chlorophyll, and it has been proved and reported in earlier studies that flavonol increases under low nitrogen availability [53,54]. This increase in flavonol ultimately causes nitrogen deficiency in plants [55]. This status of nitrogen is measured in terms of the nitrogen balance index (NB), which is also considered a very-sensitive index [56,57]. NB (chlorophyll to flavonol ratio) is a simple non-destructive metric that significantly correlates with photosynthetic photon flux density, nitrate, and nitrite levels in lettuce, making it helpful for modifying illumination intensities and predicting the efficiency of nitrate absorption in lettuce [58]. NB usually decreases under SS stress as SS directly negatively impacts chlorophyll and nitrogen levels [59]. For example, our study demonstrated that the cultivar (Ruby) with a 35.2% decrease in NB when treated with salt showed 35% loss in FM and 27.3% loss in DM. This result is further supported by the report stating that salinity can alter nitrogen metabolism in plants through the induction of specific ion effects and nutritional imbalance, thus reducing biomass yield [59]. On the other hand, some other cultivars were reported with higher FM and DM even under a lower level of NB. For example, PI 177423 showed a 23.3% loss in NB while it had a 19.3% increase in FM under SS. Thus, it can be suggested that there can be the accumulation of amides, proteins, and polyamines (which all contain nitrogen) in plants when subjected to SS, which can help protect plants from reactive oxygen species [60]. Furthermore, the response of NB towards SS was genotype-dependent.

Flavonoid and anthocyanins are also interrelated as anthocyanin belongs to a parent class called flavonoids [61]. Anthocyanins are one of the major phenols in red lettuce [62], and are induced by environmental stressors via the phenylpropanoid pathway [20]. AI usually increases under SS due to its protective function against reactive oxygen species [63]. For example, in our study, Green Forest, PI 635075, and Burpee Bibb showed a 100% increase in AI under SS. However, the FM and DM reduction were also higher (15–20%) on those cultivars. This suggests that Green Forest, PI 635075, and Burpee Bibb were more salt sensitive. Besides, AI PI 212099, and PI 171676 were unaltered by salt treatment, and they had the highest percentage increase in FM. These results suggest that the short treatment (12 days long) with salt can increase the FM and DM of lettuce without altering the AI in some salt-tolerant lettuce cultivars [64].

## 4. Materials and Methods

### 4.1. Growth Condition and Plant Materials

Thirty-eight genotypes (11 butterheads, 11 crisphead, 8 loose leaf, and 8 cos-type) were selected based on their salt tolerance and sensitivity as indicated in the literature and germplasm collection center at the U.S. Department of Agriculture, Washington. The study was completed within two months starting from Jan 2021, under greenhouse conditions at 22–25 °C daytime and 16–20 °C nighttime temperatures with a 16 h photoperiod. The lettuce plants were arranged in a randomized complete block design with four replications. There were four blocks with six replicates per treatment in each block. The greenhouse was located at the North Mississippi Research and Extension Center, Verona, MS. The temperature in the greenhouse was regulated using evaporative cooling, along with horizontal airflow fans for air circulation when needed. Lettuce seeds were planted in 96

(12 × 8) celled plastic trays filled with Pro-Mix BX Soilless medium (Premier Horticulture Inc., Quebec, QC, Canada). Pro-Mix is a mixture of sphagnum peat moss (75–85% by volume), perlite, vermiculite, limestone (for pH adjustment), and wetting agent. Twenty-one days after planting, seedlings were fertigated with 20-20-20 NPK (SQM North America Corporation, Atlanta, GA, USA) at the rate of 50 ppm.

### 4.2. Salt Treatment

Twenty-eight days after planting, sodium chloride salt treatment (NaCl) was initiated at the rate of 100 mM, which has an electrical conductivity (EC) of 9–11.8 dS/m. A 100 mM measure of NaCl was applied in split (three times) at one-day intervals to avoid excess accumulation of salt and avoid osmotic shock and sudden death of plants at higher salt concentrations. Full-strength (9–11.8 dS/m) EC of 100 mM salt treatment was reached on the fifth day of salt treatment initiation. The value of 0 mM salt in water was considered the control. Trays were treated with salt solution or normal tap water by immersing the trays with plant in the tub filled with 2.5 L of either salt solution or normal tap water and left for 10 min.

### 4.3. Morphological Traits Measurement

Forty-two days after sowing (10 days after salt treatment), six plants from each replicate per treatment were harvested, and destructive sampling was performed for the measurement of fresh mass (FM), dry mass (DM), and leaf number (LN). For DM, the leaf tissue was kept in the oven (Sheldon Manufacturing Inc., Cornelius, OR, USA) at 50 °C for 48–72 h.

### 4.4. Physiological Traits Measurement

At 42 days after sowing, the chlorophyll (CI), flavonoids (FI), anthocyanins (AI), and nitrogen balance (NB) indices were evaluated with the aid of the Dualex FORCE-A chlorophyll meter (Orsay, France) in six plants per replication per treatment, and used as described by Barickman et al. [65]. Dualex is an automatic or manual hand leaf-clip instrument that uses fluorescence and light transmission to evaluate leaf quality. It measures the 5 mm area (diameter) of leaves. The measurement reading is shown on the LCD screen with a sound warning. The CI, Fl, AI, and NB indices were measured in the ranges 0–150, 0–3, 0–3, and 0–999, respectively. NB is the ratio between CI and FI.

### 4.5. Salt Tolerance Evaluation

With some modifications, lettuce cultivars were classified using a computed salt tolerance coefficient index described by Wijewardana et al. [24]. The Salt Tolerance Coefficient Index (STCI) is the ratio of the numerical values of NaCl-treated plant/value for the control.

Total STCI was calculated using the following mathematical evaluation model:

Total STCI: LNS/LNC + FMS/FMC + NBS/NBC + CIS/CIC + AIS/AIC + FIS/FIC

Total STCI was obtained into groups (highly salt-tolerant, moderately salt-tolerant, low salt-tolerant, and highly sensitive) based on their standard deviation values.

## 5. Data Analysis

Statistical analysis of the data was performed using SAS (version 9.4; SAS Institute, Cary, NC, USA). Data were analyzed using PROC GLM analysis of variance (two-way ANOVA) followed by mean separation. The standard errors were based on the pooled error term from the ANOVA table. Tukey's test ($p \leq 0.05$) was used to differentiate between genotype classifications and treatment. Minitab (Minitab 17 Statistical Software (2010). [Computer software]. State College, PA, USA: Minitab, Inc. (www.minitab.com) (accessed on 16 July 2021) was used for principal component analysis.

## 6. Conclusions

A significant effect of SS on different morphological and physiological traits was observed in one-month-old lettuce plants. PI 212099, Buttercrunch-1, and PI 171676 were found highly salt tolerant. PI 342448, PI 615052, PI 175737, PI 593426, PI 634671, PI 612664, Parris Island, Burgundy Delight, and PI 635077 were found moderately tolerant. PI 372908, PI 358018, PI 274366, Prizehead, PI 177423, PI 289023, Iceberg, PI 635075, PI 342515, PI 358020, PI 536708, Buttercrunch-2, Crispino, PI 600773, and PI 536803 were found less salt tolerant. PI 171675, PI 536822, Ruby, PI 601339, Green Forest, PI 601488, PI 613577, PI 278066, Hybrid Bibb, PI 273606, and Burpee Bibb were found highly salt sensitive. It is interesting to note that, according to the percentage of growth decline, cultivars with high growth potential were significantly salt-sensitive, whereas cultivars with high salt tolerance usually had poor growth potential under control conditions. This suggests that the morphological and physiological response of 38 lettuce cultivars towards SS is genotype-dependent. All above demonstrations and reports will be a basis for understanding the effect of salt concentration in different types of lettuce cultivars with varying levels of salt tolerance. Furthermore, the study will help identify physiological, morphological, and biochemical attributes in lettuce due to SS that may help plant breeders develop salt-tolerant lettuce genotypes.

**Author Contributions:** B.A.: conceptualization, methodology, validation, formal analysis, investigation, writing—original draft, writing—review and editing, visualization, methodology, validation, investigation. O.J.O.: methodology, validation, formal analysis, investigation. J.C.W.: methodology, validation, formal analysis, investigation, writing—review and editing. T.C.B.: conceptualization, methodology, validation, formal analysis, investigation, resources, data curation, writing—review and editing, visualization, supervision, project administration, funding acquisition. All authors have read and agreed to the published version of the manuscript.

**Funding:** This material is based on the work supported by the USDA-NIFA Hatch Project under accession number 149210.

**Institutional Review Board Statement:** Not applicable.

**Informed Consent Statement:** Not applicable.

**Data Availability Statement:** The data presented in this study are available on request from the corresponding author.

**Acknowledgments:** We thank Thomas Horgan and Skyler Brazel at the Vegetable Physiology Laboratory for their technical assistance for their help during data collection.

**Conflicts of Interest:** The authors declare that they have no known competing financial interest or personal relationships that could have appeared to influence the work reported in this paper.

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
