# Peer review of "Morphological and Physiological Response of Different Lettuce Genotypes to Salt Stress"

_stresses, doi:10.3390/stresses1040021_

Round 1

Reviewer 1 Report

The manuscript reported Salt stress (SS) on lettuce. There are a few issues to be addressed. Is this the common lettuce plant used in the study? It is understood high salt concentration affects absorption by the plant not only through the root. What is point of the following statement,".. making it difficult to extract the soil’s water and essential mineral nutrients." In addition, the sample analysis needs to be elaborated in detail to demonstrate the significance of the findings. Finally, more updated references on the subject matter should be cited.

Author Response

Please see attached response to reviewers document. Thank you for your service!

Reviewer 2 Report

Why are there no Materials & Methods? See attached for some specific comments. So, I put major revision. Otherwise, the paper is mostly well-written. A bit wordy in my opinion but it's ok. Also, please include some of the data in graphic form, it would make the paper easier to read. All those numbers and letters and mind-numbing.

Author Response

(The authors gave the same response as above.)

Reviewer 3 Report

Although the article includes quite a fair amount of experimental data, the manuscript has some weaknesses that make it unpublishable. First, the data are not novel and the way the results are presented could be improved. I have not seen the description of material and methods that in the journal Stresses is found in section 4 before Conclusions. 
I have detected many errors in the way ideas are expressed. For example, it seems inappropriate to speak of highly salt tolerant lettuce cultivars, and many other inappropriate expressions or misleading statements .

Author Response

(The authors gave the same response as above.)

Round 2

Reviewer 1 Report

The revised manuscript reads better.

Author Response

Please see the attached response to reviewers comments. Thank you!

Reviewer 2 Report

The manuscript is much improved and merits publication in its present form.

Author Response

(The authors gave the same response as above.)

Reviewer 3 Report

The authors have included the Material and method section, but the rest of the text has only been slightly modified, although I have recommended deleting unclear and misleading statements, such as" Oxidative stress mainly affects plant growth and metabolism, leading to nutritional imbalance, osmotic stress and water deficit (L-61-62) (when it is the other way around! )-, or that they refer to highly salt-tolerant lettuce genotypes (instead of relatively salt-tolerant genotypes, since lettuce is obviously a salt-susceptible glycophyte)!
I have more specific comments: The ideas in the Introduction do not follow a logical order. 
Material and methods section needs more clarity.  A concentration of 100mM salt is given - what salt?
How many replicates per culture per treatment? 
The duration of salt treatments and the manner of application is also unclear.  
Results
The final EC of the substrate is not indicated, although it was measured.  
The number of replicates for each cultivar and treatment is not clearly specified in the tables. In the table headings the authors refer to 42 days after sowing, when what is relevant is the duration of the salt stress treatment. 
The statistical treatment is not optimal. Instead of a one-way ANOVA, a two-way Anova would be more appropriate. A graphical representation of the parameters showing a significant interaction would be useful....
The PCA is not well explained; the PCA "was also performed for all morphological and physiological traits for 38 genotypes. 
". Where?  The technical quality of the graph is low. 
I would recommend the introduction of a loading plot of the principal component analysis (PCA) performed on the traits analyzed, and separately a scatter plot of the PCA scores.
The Discussion is a simple repetition of the results. 

Author Response

(The authors gave the same response as above.)

Round 3

Reviewer 3 Report

The manuscript is much  improved in this new version. However, authors should better explain the watering procedure, which remains unclear.